# Signal Thresholding Segmentation of Ventricular Volumes in Young Patients with Various Diseases—Can We Trust the Numbers?

**DOI:** 10.3390/diagnostics13020180

**Published:** 2023-01-04

**Authors:** Titus Thut, Emanuela Valsangiacomo Büchel, Julia Geiger, Christian Johannes Kellenberger, Beate Rücker, Barbara Elisabeth Ursula Burkhardt

**Affiliations:** 1Pediatric Cardiology, Department of Surgery, Pediatric Heart Center, University Children’s Hospital Zurich, 8032 Zurich, Switzerland; 2Children’s Research Center, University Children’s Hospital Zurich, 3032 Zurich, Switzerland; 3Department of Diagnostic Imaging, University Children’s Hospital Zurich, 8032 Zurich, Switzerland

**Keywords:** cardiac magnetic resonance imaging, ventricular volume, segmentation, automation, flow

## Abstract

In many cardiac diseases, right and left ventricular volumes in systole and diastole are diagnostically and prognostically relevant. Measurements are made by segmentation of the myocardial borders on cardiac magnetic resonance (CMR) images. Automatic detection of myocardial contours is possible by signal thresholding techniques, but must be validated before use in clinical settings. Biventricular volumes were measured in end-diastole (EDVi) and in end-systole (ESVi) both manually and with the MassK application, with signal thresholds at 30%, 50%, and 70%. Stroke volumes (SV) and cardiac indices (CI) were calculated from volumetric measurements and from flow measured in the ascending aorta and the main pulmonary artery, and both methods were compared. Reproducibility of volumetric measurements was tested in 20 patients. Measurements were acquired in 94 patients aged 15 ± 9 years referred for various conditions. EDVi and ESVi of both ventricles were largest with manual segmentation and inversely proportional to the MassK threshold. Manual and k30 SV and CI corresponded best to flow measurements. Interobserver variability was low for all volumes manually and with MassK. In conclusion, manual and 30% threshold-based biventricular volume segmentation agree best with two-dimensional, phantom-corrected phase contrast flow measurements in a young cardiac referral population and are well reproducible.

## 1. Introduction

Diagnosis and prognosis in many cardiovascular diseases rely on the accurate quantification of left and right ventricular volumes as well as myocardial mass measurements [1,2,3,4,5,6]. Cardiovascular magnetic resonance (CMR) imaging is currently considered the reference standard for the measurement of ventricular volume and function [7,8]. Traditionally, calculation of volumes and mass is based on manual segmentation of the endocardial and epicardial borders or automatic border detection followed by manual correction. This approach is time-consuming and hides some inaccuracies, as trabeculations and papillary muscles (TPM) result to be part of the blood volume [9]. This is generally accepted, because correct manual tracing of TPM is challenging and may increase intra- and inter-observer variability [10,11]. Nevertheless, in recent studies, TPM has been shown to have a significant effect on the correct quantification of ventricular volume and mass [10,11,12,13]. Inclusion of TPM into the left ventricular (LV) cavity was shown to lead to overestimation of end diastolic volume (EDV) and end systolic volume (ESV) and underestimation of left ventricular mass (LVM) [10,14]. This is relevant in specific diseases such as hypertensive and hypertrophic cardiomyopathy [12] or LV non-compaction, where ventricular volume and mass are part of the diagnostic criteria [13,15,16].

Newer MR post-processing algorithms use automatic myocardial contour detection based on signal thresholding. This technique improves discrimination between myocardium and blood, is less operator-dependent and reduces post-processing time [17,18]. However, reproducibility and accuracy of a threshold-based, semi-automated segmentation algorithm must be evaluated in comparison with manual contour-based segmentation and flow measurements as independent reference methods, before they can be used interchangeably for clinical care.

Our aim was to assess the accuracy of semi-automated CMR imaging segmentation measurements of ventricular volumes and mass, as well as of calculation of stroke volume (SV), and cardiac index (CI) for the LV and the right ventricle (RV). We aimed to validate these parameters against both ventricular manual contour drawing measurements and phantom corrected 2D flow measurements and to test reproducibility.

## 2. Materials and Methods

### 2.1. Patient Selection

We retrospectively reviewed consecutive data acquired in our institution between January 2018 and March 2019. Inclusion criteria consisted of biventricular cardiac physiology, MR studies containing steady-state free precession (SSFP) images for ventricular function and two-dimensional phase contrast (2D PC) flow sequences. Exclusion criteria were defined as: incomplete MR data; lack of consent for retrospective data analysis; significant stenosis or regurgitation in one or more cardiac valves shown by 2D PC flow or by echocardiography; implantable cardiac device; major arrhythmia during the scan.

The ethics committee of the canton of Zurich, Switzerland (KEK-Nr. 2019-00422) approved the conduct of this study, and all patients, their parents or legal guardians, gave informed consent to the use of their data.

### 2.2. Imaging Technique

All CMR studies were performed with a 1.5 Tesla scanner (Signa HDx, GE Medical Systems, Milwaukee, WI, USA). Subjects were imaged in supine position during breath-hold at end-expiration. SSFP images were acquired in a stack of short axis slices covering both ventricles with the following parameters: 40 cardiac phases, 10–13 slices as appropriate, slice thickness 6–8 mm depending on body size, interslice gap 0–2 mm, temporal resolution <30 ms, echo time 1.5–1.8 ms, repetition time 3.5–4.2 ms, flip angle 45°, matrix size 224 × 224, field of view 280–420 mm. Vertical and horizontal long axis planes were used for correct perpendicular planning of the short axis images.

Phantom-corrected 2D PC images were acquired for flow measurements perpendicular to the ascending aorta at the level of the right pulmonary artery and in the main pulmonary artery (MPA) midway between the pulmonary valve and the bifurcation, this was used as an independent reference standard [19]. Acquisition parameters were: field of view 240–400 mm, slice thickness 4 mm, matrix 256 × 128, 20 phases, views per segment 6–10, echo time 1.18–1.35 ms, repetition time 2.75–3.06 ms, flip angle 20°, and velocity encoding 200 cm/s.

### 2.3. Image Analysis

Image quality was assessed visually. All ventricular volumes were measured twice; once using the contour-based method and once using the threshold-based method. To avoid bias, both measurements were performed by the same operator in a time interval of more than two weeks.

### 2.4. Ventricular Volumes

Manual contour-based segmentation was performed on SSFP short axis images using QMass (version 8.1, Medis, Leiden, The Netherlands) as previously reported [6,20]. TPM were included as part of the blood pool. The interventricular septum was considered part of the left ventricle.

Threshold-based volumes were calculated on the same SSFP images with the MassK thresholding algorithm of the same software. The same end-systolic and end-diastolic phases and the same epicardial contours defined manually were utilized for MassK application. The MassK algorithm discriminates blood from myocardium by analysis of voxel signal intensity as described by Jasper et al. [18]. Measurements were taken with signal intensity thresholds set at 30% (k30), 50% (k50), and 70% (k70), respectively. Figure 1 shows one case of manual segmentation and threshold-based segmentation.

### 2.5. Flow Volumes

Velocities and flow volumes were calculated using QFlow software (version 8.1, Medis, Leiden, The Netherlands). All 20 phases were contoured by using a semi-automated algorithm followed by manual correction where necessary. Offset errors were corrected using phantom acquisitions [21]. The SV obtained in the aorta and in the MPA (mL × heart rate/body surface area [m^2^]) were used as references for ventricular SV and CI calculated by both methods.

### 2.6. Interobserver Variability

Interobserver variability was tested by comparing manual contours and threshold-based contours performed by two independent observers of different experience grade (TT and BB) in 20 patients.

### 2.7. Statistical Analysis

Statistical analysis was performed using GraphPad Prism version 8.4.3 for OS X (GraphPad Software, San Diego, CA, USA). D’Agostino-Pearson omnibus normality test was used to check distribution of data. Mean±standard deviation was used to present normally distributed values and median ± interquartile range for non-normally distributed values. LV and RV volume indices (volume/body surface area (BSA) [mL/m^2^]) and mass indices (mass/BSA [g/m^2^]) using manual segmentation (QMass) and semi-automated segmentation (MassK) were compared using paired t-tests and Pearson’s correlation coefficients for data with normal distribution, or Wilcoxon tests and Spearman’s correlation coefficients for non-normally distributed data, respectively. Patient groups were compared using the Mann–Whitney test. Bland–Altman analysis was used to assess agreement between the different segmentation techniques and interobserver variability [22]. Interobserver variability was also evaluated by intraclass correlation coefficient analysis.

Coefficient of variation was calculated as standard deviation of a measurement divided by the mean value. *P* values < 0.05 were considered statistically significant.

## 3. Results

### 3.1. Patient Characteristics

A total of 94 patients (22 female) fulfilled the inclusion criteria. Details about demographics, patient characteristics, and diagnoses are described in Table 1. Cardiomyopathy included muscular dystrophy (*n* = 17), Kawasaki disease (*n* = 6), hypertrophic cardiomyopathy (*n* = 3), Marfan disease (*n* = 2), metabolic disease (*n* = 1), and noncompaction cardiomyopathy (*n* = 1). Congenital heart disease included corrected dextro-transposition of the great arteries (*n* = 13), total anomalous pulmonary venous return (*n* = 4), partial anomalous pulmonary venous return (*n* = 3), ventricular septal defect (*n* = 2), anomalous left coronary artery origin from the pulmonary artery (*n* = 1), hypoplastic pulmonary arteries (*n* = 1), double outlet RV (*n* = 1) and atrioventricular septal defect (*n* = 1). Aortic pathology included coarctation of the aorta (*n* = 9), bicuspid aortic valve (*n* = 6), interrupted aortic arch (*n* = 3), vascular ring (*n* = 2), aortic dissection (*n* = 1), aortic aneurysm (*n* = 1), and double aortic arch (*n* = 1). Healthy subjects underwent CMR for screening purposes but showed normal cardiac anatomy and function. Others were LV fibroma (*n* = 1) and LV carcinoma (*n* = 1).

### 3.2. Ventricular Volumes

Ventricular volumes calculated by manual segmentation and by threshold-based contouring with 3 different threshold values (k30, k50, and k70) as well as flow volumes measured in the aorta and MPA are reported in Table 2.

As expected, the largest volumes (EDV and ESV) were measured with manual segmentation in both ventricles. Volumes decreased steadily with increasing thresholds. At all threshold settings used, LV EDVi and LV ESVi were significantly smaller than by manual measurements, respectively (*p* < 0.0001 for all). Additionally, RV EDVi and RV ESVi were significantly smaller than by manual measurements (*p* < 0.0001 for all).

Similarly, myocardial mass showed the lowest values with manual segmentation and increased with increasing thresholds. At a threshold of 50%, LVMi and RVMi were significantly greater than manual measurements (*p* < 0.0001 for both).

In contrast, ejection fraction was not influenced by the technique of segmentation.

### 3.3. SV and CI Comparison

LV SV was measured significantly different compared to the flow in the aorta with all segmentation techniques except for the k30 threshold-based measurement. LV CI did not differ from flow for manual segmentation and k30 threshold. In contrast, k50 and k70 thresholds presented with a systematic underestimation of SV and CI as shown by Bland–Altman analysis (Table 3).

RV SV and CI were significantly different from flow measures in the MPA. A systematic underestimation was found for SV but not for CI (Table 3).

There was a greater discrepancy between manual and flow-derived stroke volumes in patients with (manual) LVEF above median versus below median (5.2 vs. −1.1 mL; *p* = 0.001), and between patients with (manual) RVEF below median versus above median (−9.0 vs. 1.9 mL; *p* < 0.001).

The discrepancy between manual stroke volumes and flow-derived stroke volumes of the respective ventricle was not statistically different between patients with (manual) LVEDVi below median versus above median (*p* = 0.147) or with (manual) RVEDVi below versus above median (*p* = 0.368).

Similarly, the discrepancy between k30 stroke volumes and flow-derived stroke volumes was greater with higher versus lower (manual) LVEF (3.1 vs. −1.0 mL; *p* = 0.012) but was greater with lower versus higher (manual) RVEF (−9.5 vs. 0.7 mL; *p* < 0.001) and lower versus higher (manual) RVEDVi (−7.3 vs. −1.7 mL; *p* = 0.039), with no statistical significance between LVEDVi strata (*p* = 0.057) (Figure 2 and Figure 3).

### 3.4. Variability

Table 4 shows high coefficients of variation for the respective measurements, reflective of our diverse patient population, but similar between volumetric methods.

### 3.5. Reproducibility

Bland–Altman analysis for all volume indices (LV EDVi, LV ESVi, RV EDVi, RV ESVi) by threshold-based (30%, 50% and 70%) and manual measurements showed acceptably small biases between the two readers (bias from −0.4 to −4.8 mL/m^2^). All CI measurements (LV and RV) with manual, all threshold-based methods, and flow showed small biases between the two readers (bias from 0 to 0.2 l/min/m^2^, Table 5). Similar outcomes were measured for SV (bias from −0.1 to 3.8 mL). The smallest biases were found in the flow measurements (bias of 0 l/min/m^2^ for CI, similarly −0.1 and −0.2 mL for SV). In readings for RV volumes, the threshold-based method showed overall smaller biases when compared to the manual method (Table 5).

## 4. Discussion

This study was performed to evaluate the accuracy and precision in the assessment of ventricular volumes from CMR images in a clinical referral cohort using threshold-based contour detection. We assessed both the accuracy of the measurements made with different threshold settings compared to phantom-corrected [23] flow volumes, as well as their precision in terms of interobserver reproducibility.

We observed that the threshold-based tracings resulted in significantly smaller EDVi and ESVi of both ventricles than the manual method. VMi for both ventricles was significantly higher by the threshold-based method than by the manual method. This can be explained by the TPM, which was included in the threshold-based but not in the manual method, and which can constitute up to 23% of EDV and 28% of LVM [13].

In addition, we assessed the accuracy of all volumetric methods compared to flow data. In the LV, the threshold-based method at a signal intensity of 30% was closest to the reference standard, while thresholds of 50% and 70% underestimated LV CI. For RV CI, all investigated methods showed significant deviation from flow measurements, with manual and the 30% threshold method being closest to the reference standard. For this reason, a low threshold such as 30% is preferable to higher thresholds as a substitute for fully manual segmentation. The user must deliberately change the software’s standard threshold of 50% for each examination, if not even adjusting the threshold slice by slice.

Previous studies have compared different automated and semi-automated methods to accurately and time-efficiently quantify LV volumes and mass in adults [11]. Similarly to our study, Varga-Szemes et al. have studied the MassK algorithm [17] and found that SV from threshold-based and flow-based measurements were in agreement but were significantly different from the manual method for the LV. Jaspers et al. [18] also studied the LV of fewer subjects with MassK, all with normal cardiac function and anatomy, and found even higher differences between methods than the former. Interobserver agreement of this threshold-based algorithm was higher than with manual contouring for both ventricles in a healthy cohort of 60 young adults [24].

There is one report in patients with repaired tetralogy of Fallot with lower RV volumes and higher RVEF and mass by threshold-based compared to manual contours, with better reproducibility by the threshold-based method for RV mass, but not for RV volumes or EF [25]. Patients with a systemic RV after Senning procedure with hypertrophy of the wall and trabeculations have been examined before, with better correlation of RV stroke volumes and flow using MassK than using manual contours [26].

This present investigation is unique as it studies a younger population (15 ± 9 vs. 55 ± 18 years old) with a broad variety of different diagnoses reflective of part of a typical clinical referral population. In addition, we evaluated not only LV, but also RV volume and mass measurements, which are important in many congenital heart diseases [27].

Our study showed better interobserver agreement for RV volumes with the threshold-based compared to the manual method. Excellent interobserver agreement was found for biventricular EDVi and CI. Of all parameters, RV ESVi and the mass indices of both ventricles showed the highest interobserver variability by all methods, which exemplifies the importance of manual epicardial tracing, including in the RV outflow tract. This is a potential source of error that cannot be overcome by the semi-automated thresholding technique. At the same time, RV CI showed the least bias from flow measurements by manual volumetrics.

To our knowledge, we provide the first evaluation of biventricular threshold-based volumetrics in a varied, clinical referral population of young patients with different congenital and acquired heart diseases using phantom-based flow as a reference standard. Patients with valvular dysfunction were excluded in order to avoid flow alterations and to use only valid flow data [28]. We demonstrated that a high accuracy of right and left ventricular volume measurements is achievable with certain threshold settings. Precision (coefficients of variation) did not improve using the semi-automated threshold-based method compared to manual contouring. However, both methods performed well when employed by observers with different levels of experience. Hence, threshold-based ventricular volumetrics may be attractive for centers with multiple readers providing speed and consistency even for repetitive follow up examinations.

Limitations of this study are its retrospective nature and a small patient population, although greater than in previous published studies. Our results may not be generalizable to healthy individuals, as has been done before by others [29], because we studied a clinical referral population. Results of the threshold-based measurements from this study should also not be used as normal values, because only 14% of our population were healthy subjects. We are not reporting analysis times, since the focus of our study was assessment of accuracy and precision more than efficacy. Nevertheless, even with manual drawing of epicardial contours and automatic addition of endocardial contours by the MassK algorithm, still only approximately half the time would be required than for contouring both epicardial and endocardial contours. Previous studies of the same and similar algorithms have proven significant time reduction [13,17,30].

## 5. Conclusions

In conclusion, our results show sufficient accuracy of the MassK threshold-based algorithm for the evaluation of LV and RV volumes if the signal intensity threshold is adjusted to 30%. We demonstrate that the threshold-based algorithm is also suitable for young patients with various cardiac pathologies. Excellent agreement was found between observers with different experience levels. Thus, the threshold-based method can be used interchangeably with the conventional manual method to increase efficiency in clinical care, if the user adapts the threshold setting correctly.

## Figures and Tables

**Figure 1 diagnostics-13-00180-f001:**
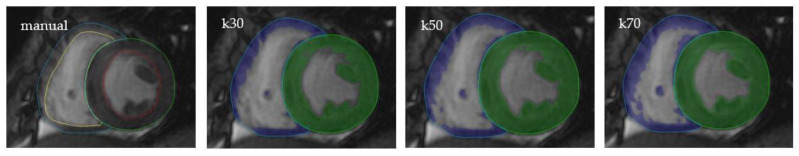
Representative examples of the manual measurement technique and the threshold-based measurement techniques at 30%, 50%, and 70%, respectively (k30, k50, and k70), in a mid-ventricular end-systolic frame. Manual contour detection clearly excludes the trabeculations and papillary muscle from the myocardium, while threshold-based segmentation includes these structures. The higher the threshold is set, the more myocardium is detected.

**Figure 2 diagnostics-13-00180-f002:**
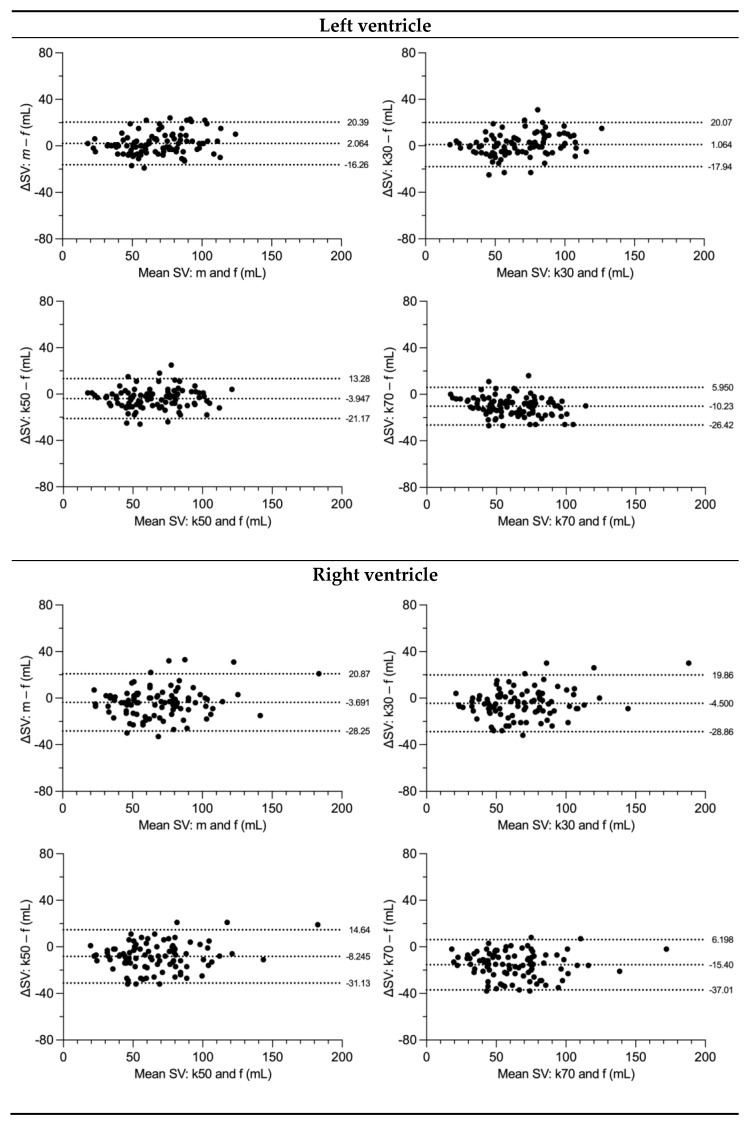
Bland–Altman (B.-A.) plots comparing stroke volumes of flow measurements with manual and threshold-based methods. Flow was measured as stroke volume in the ascending aorta for the left ventricle and as stroke volume in the main pulmonary artery for the right ventricle. Dotted lines show the mean of differences and the 95% limits of agreement (±1.96 SD), ΔSV difference in stroke volume, SD standard deviation from mean, f flow measurement, m manual measurement, k30 threshold-based measurement at 30% signal intensity, k50 threshold-based measurement at 50% signal intensity, k70 threshold-based measurement at 70% signal intensity.

**Figure 3 diagnostics-13-00180-f003:**
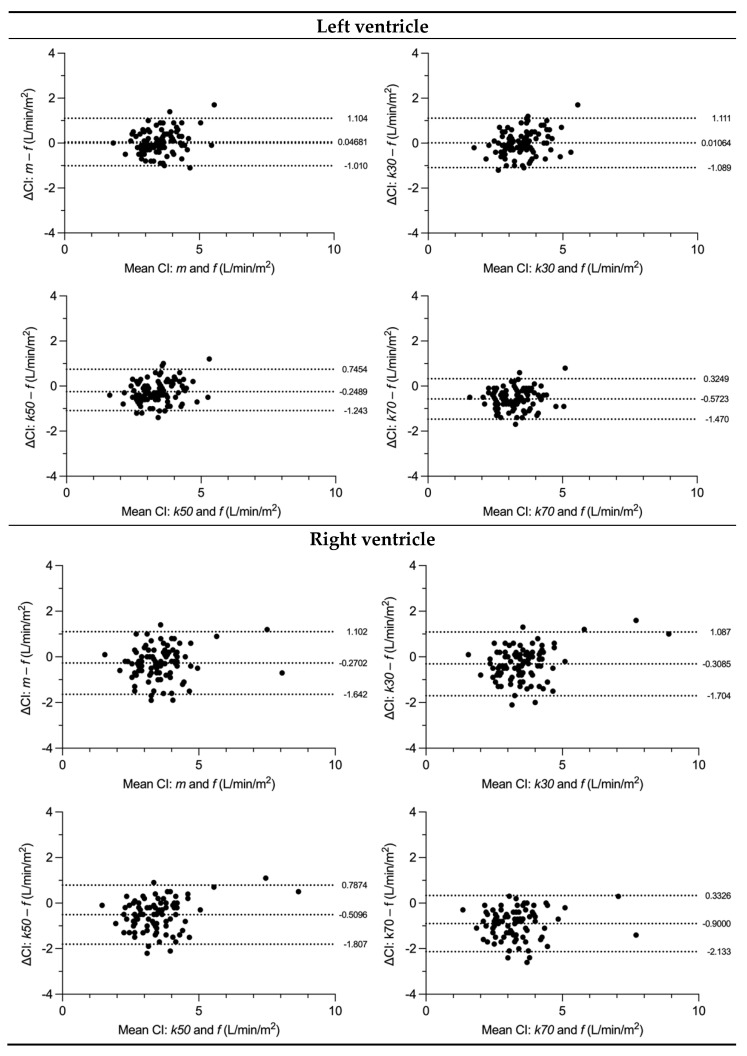
Bland–Altman (B.-A.) plots comparing cardiac index of flow measurements with manual and threshold-based methods. Flow was measured as cardiac index in the ascending aorta for the left ventricle and as cardiac index in the main pulmonary artery for the right ventricle. Dotted lines show the mean of differences and the 95% limits of agreement (±1.96 SD), ΔCI difference in cardiac index, SD standard deviation from mean, f flow measurement, m manual measurement, k30 threshold-based measurement at 30% signal intensity, k50 threshold-based measurement at 50% signal intensity, k70 threshold-based measurement at 70% signal intensity.

**Table 1 diagnostics-13-00180-t001:** Patient characteristics (*n* = 94).

Age (years)	15 ± 9
Weight (kg)	49.9 ± 20.5
Height (cm)	154 ± 22
Body surface area (m^2^)	1.4 ± 0.4
Heart rate (bpm)	77 ± 14
Diagnosis	
Cardiomyopathy	30 (32%)
Congenital heart disease	26 (28%)
Aortic pathology	23 (24%)
Healthy subject	13 (14%)
Other	2 (2%)

Values are shown as mean ± standard deviation or numbers (percentage).

**Table 2 diagnostics-13-00180-t002:** Volumes and flow measured manually and with threshold-based contouring.

	Manual	k30	k50	k70	Flow
Left ventricle					
EDVi (mL/m^2^)	82.3 ± 16.1	75.4 ± 16.0	67.2 ± 14.4	59.6 ± 13.0	N/A
ESVi (mL/m^2^)	35.1 ± 11.0	28.8 ± 10.4	24.1 ± 8.9	20.7 ± 7.9	N/A
EF (%)	60 ± 10	60 ± 10	60 ± 10	70 ± 10	N/A
SV (mL)	68.8 ± 24.9	67.8 ± 25.0	62.8 ± 23.1	56.5 ± 20.6	66.8 ± 22.7
CI (l/min/m^2^)	3.6 ± 0.8	3.5 ± 0.8	3.3 ± 0.8	2.9 ± 0.7	3.5 ± 0.7
Mass (g/m^2^)	54.4 ± 11.0	60.8 ± 12.5	69.1 ± 13.8	76.3 ± 16.6	N/A
Right ventricle					
EDVi (mL/m^2^)	86.6 ± 22.2	80.1 ± 22.1	72.4 ± 20.9	62.7 ± 18.8	N/A
ESVi (mL/m^2^)	41.0 ± 13.9	35.0 ± 12.3	29.9 ± 11.1	25.3 ± 9.7	N/A
EF (%)	50 ± 10	60 ± 10	60 ± 10	60 ± 10	N/A
SV (mL)	66.6 ± 28.6	65.8 ± 29.6	62.1 ± 28.5	54.9 ± 26.2	70.3 ± 26.6
CI (l/min/m^2^)	3.4 ± 1.0	3.5 ± 1.2	3.2 ± 1.1	2.8 ± 1.0	3.7 ± 0.9
Mass (g/m^2^)	21.8 ± 5.2	28.6 ± 6.5	36.6 ± 8.3	46.8 ± 10.6	N/A

Values are shown as mean ± standard deviation. k30 threshold-based measurement at 30% signal intensity, k50 threshold-based measurement at 50% signal intensity, k70 threshold-based measurement at 70% signal intensity, N/A not applicable, EDVi end diastolic volume index, ESVi end systolic volume index, SV stroke volume, CI cardiac index, Mass ventricular mass index.

**Table 3 diagnostics-13-00180-t003:** Comparison of manual and different threshold-based stroke volume and cardiac index versus flow.

Bland–Altman of	Manual	k30	k50	k70
Left ventricle SV (mL)	2.1 ± 9.4	1.1 ± 9.7	−4.0 ± 8.8	−10.2 ± 8.3
Right ventricle SV (mL)	−3.7 ± 12.5	−4.5 ± 12.4	−8.3 ± 11.7	−15.4 ± 11.0
Left ventricle CI (l/min/m^2^)	0.1 ± 0.5	0.0 ± 0.6	−0.3 ± 0.5	−0.6 ± 0.5
Right ventricle CI (l/min/m^2^)	−0.3 ± 0.7	−0.3 ± 0.7	−0.5 ± 0.7	−0.9 ± 0.6

Bland–Altman analysis is depicted as bias ± 1.96 standard deviations. SV stroke volume, CI cardiac index, k30 threshold-based measurement at 30% signal intensity, k50 threshold-based measurement at 50% signal intensity, k70 threshold-based measurement at 70% signal intensity.

**Table 4 diagnostics-13-00180-t004:** Coefficient of variation.

	Manual	k30	k50	k70	Flow
Left ventricle					
EDVi (mL/m^2^)	19.5%	21.2%	21.4%	21.8%	N/A
ESVi (mL/m^2^)	31.4%	36.1%	37.1%	38.1%	N/A
EF (%)	13.6%	13.7%	13%	12.7%	N/A
SV (mL)	36.1%	36.9%	37.8%	36.4%	34%
CI (l/min/m^2^)	21.2%	23.6%	25%	24.1%	18.9%
Mass (g/m^2^)	20.5%	20.5%	23.7%	21.8%	N/A
Right ventricle					
EDVi (mL/m^2^)	25.7%	27.6%	28.9%	29.9%	N/A
ESVi (mL/m^2^)	34%	35.2%	37%	38.2%	N/A
EF (%)	16.1%	15.5%	15%	15%	N/A
SV (mL)	42.9%	44.9%	45.9%	47.6%	37.8%
CI (l/min/m^2^)	29.6%	34.2%	34.9%	35%	25.4%
Mass (g/m^2^)	24%	22.9%	22.6%	22.7%	N/A

Coefficient of variation equals the standard deviation divided by the mean. k30 threshold-based measurement at 30% signal intensity, k50 threshold-based measurement at 50% signal intensity, k70 threshold-based measurement at 70% signal intensity, N/A not applicable, EDVi end diastolic volume index, ESVi end systolic volume index, SV stroke volume, CI cardiac index, Mass ventricular mass index.

**Table 5 diagnostics-13-00180-t005:** Interobserver analysis between two readers.

	Manual	k30	k50	k70	Flow
B.-A.					
Left ventricular parameters					
EDVi (mL/m^2^)	−1.0 ± 3.1	1.1 ± 2.8	1.3 ± 2.5	1.4 ± 2.1	N/A
ESVi (mL/m^2^)	−0.6 ± 4.1	−1 ± 3.5	−0.6 ± 3.1	−0.4 ± 2.8	N/A
SV (mL)	−0.5 ± 7.2	3.3 ± 6.8	2.9 ± 6.3	2.7 ± 5.3	−0.1 ± 0.2
CI (l/min/m^2^)	0 ± 0.3	0.2 ± 0.4	0.2 ± 0.3	0.1 ± 0.3	0 ± 0
VMi (g/m^2^)	−2.2 ± 3.9	−4.5 ± 2.8	−4.5 ± 3	−4.4 ± 3.3	N/A
Right ventricular parameters					
EDVi (mL/m^2^)	−2.4 ± 3.7	−0.7 ± 3.3	−0.6 ± 3.0	−0.6 ± 3.0	N/A
ESVi (mL/m^2^)	−4.79 ± 5.41	−3.34 ± 4.19	−2.48 ± 3.77	−2.0 ± 3.4	N/A
SV (mL)	3.40 ± 7.42	3.75 ± 6.42	2.65 ± 6.02	2.0 ± 5.4	−0.15 ± 0.67
CI (l/min/m^2^)	0.17 ± 0.39	0.22 ± 0.34	0.14 ± 0.30	0.1 ± 0.3	−0.02 ± 0.07
VMi (g/m^2^)	−1.87 ± 2.23	−3.48 ± 2.25	−3.68 ± 2.50	−3.6 ± 2.5	N/A
*ICC*					
Left ventricular parameters					
EDVi	0.99	0.99	0.99	0.99	N/A
ESVi	0.96	0.96	0.96	0.96	N/A
SV	0.94	0.96	0.96	0.96	1.00
CI	0.87	0.90	0.91	0.92	1.00
VMi	0.91	0.97	0.97	0.97	N/A
Right ventricular parameters					
EDVi	0.99	0.99	0.99	0.99	N/A
ESVi	0.91	0.93	0.93	0.93	N/A
SV	0.96	0.974	0.98	0.98	1.00
CI	0.88	0.922	0.94	0.94	1.00
VMi	0.88	0.95	0.96	0.97	N/A

Values are shown as mean ± standard deviation or ICC value. k30 threshold-based measurement at 30% signal intensity, k50 threshold-based measurement at 50% signal intensity, k70 threshold-based measurement at 70% signal intensity, N/A not applicable, EDVi end diastolic volume index, ESVi end systolic volume index, SV stroke volume, CI cardiac index, VMi ventricular mass index, ICC intraclass correlation coefficient.

## Data Availability

Not applicable.

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
