# Peer review of "Signal Thresholding Segmentation of Ventricular Volumes in Young Patients with Various Diseases—Can We Trust the Numbers?"

_diagnostics, 2023, doi:10.3390/diagnostics13020180_

Round 1

Reviewer 1 Report

Have you tried to compare the manual trace including the papillary muscle? 

Author Response

Have you tried to compare the manual trace including the papillary muscle? 

>No, we didn’t trace the papillary muscles this time. It would have been interesting to see if manual tracing of the papillary muscle would have changed the ventricular volumes, but maybe not the stroke volumes, because that would have affected both diastole and systole. Thank you for your suggestion.

Reviewer 2 Report

Thut et al invesigated the accuracy of the MassK threshold-based gorithm for the evaluation of LV and RV volumes compared to manual contouring in young patients with various cardiac pathologies. The manuscript was well-written, however, i have some comments that should be concerned.

1.     I wonder if patients with arrhythmia were included in the study.

2.     Reproducibility include intraobserver and interobserver reproducibility. Similarly, variability also include intraobserver and interobserver Reproducibility. So, please add the intraobserver variability and reproducibility.

3.     In figures 2 and 3, the values indicating bias and the 95% limits of agreement were missing in the right panels.

4.     Did the biventricular sizes (normal or dilated cardiac chamber) have an impact on the accuracy of threshold-based method compared to manual contouring? Likewise, did normal or decreased biventricular EF have an influence on the accuracy of threshold-based method? Please add the supported data and results.

Author Response

Thut et al invesigated the accuracy of the MassK threshold-based gorithm for the evaluation of LV and RV volumes compared to manual contouring in young patients with various cardiac pathologies. The manuscript was well-written, however, i have some comments that should be concerned.

  1. I wonder if patients with arrhythmia were included in the study.

>No, there were no patients with arrhythmia during the study. The common respiratory arrhythmia which affects patients in sinus rhythm was taken into account using arrhythmia rejection in both the SSFP and in the phase contrast sequences. Arrhythmia was now added to the paragraph about exclusion criteria.

  1. Reproducibility include intraobserver and interobserver reproducibility. Similarly, variability also include intraobserver and interobserver Reproducibility. So, please add the intraobserver variability and reproducibility.

> Formally, reproducibility can be tested between different observers and between measurements of the same observer, but here, only interobserver reproducibility was tested and not intraobserver reproducibility. Even the comparison between two different readers showed very small biases (table 5), so it would be unlikely to find more relevant biases in intraobserver comparisons. It would take much more time to perform the additional measurements, unfortunately, if we were to add intraobserver data as well.

  1. In figures 2 and 3, the values indicating bias and the 95% limits of agreement were missing in the right panels.

> Sorry, this was due to a formatting problem with the page margin, and they are visible now.

  1. Did the biventricular sizes (normal or dilated cardiac chamber) have an impact on the accuracy of threshold-based method compared to manual contouring? Likewise, did normal or decreased biventricular EF have an influence on the accuracy of threshold-based method? Please add the supported data and results.

> This additional question was examined and the results added in paragraph 3.3

Reviewer 3 Report

This is well-presented, interesting and important analysis performed in relatively poor examined group of young patients with predominant congenital heart diseases. I find it adequate for publication in its present form.

-The main question is the assessment and comparison of threshold based myocardial detection with different threshold included in MRI for the evaluation of left ventricular and right ventricular volumes. As the reference method for stroke volume and cardiac indices volumetric and flow measurements at the level of ascending aorta and pulmonary artery were used. Additionally, biventricular volumes were calculated with manual tracing of myocardial border in short axis view for both the left and right ventricle. This last method by definition includes TPM (trabeculations and papillary muscles) into ventricular volumes contrary to threshold based techniques.

-In my opinion the topic is relevant because the analysis encompassed the relatively poorly examined populations of pediatric and adolescent patients mainly with congenital heart diseases, which included 94 subjects. Tested in the study thresholding segmentation method has a potential of more accurate quantification of myocardial mass which may improve the stratification of hypertrophy, especially when the TPM forms the significant percentage of the LV/RV mass.

-Automatic myocardial contour detection based on threshold segmentation has a potential to improve the accuracy of full myocardial mass (including TPM) and real chambers volume assessment. Moreover, seems to be faster and less operator-dependent than manual tracing of epicardial and endocardial border. Recently, some authors have published the studies aiming at description of normal values (in relation to age and sex data) for this relatively new technique in the calculation of LV and RV chambers as well as the myocardial mass, mainly in medium sized group of healthy subjects. The novelty of the submitted study concerns the study group examined, LV and RV evaluation, as well as the analysis and comparison of three different threshold cut-offs for optimal detection and quantification of the myocardial tissue.

-In my opinion the methodology is described in detailed and clear manner and does not need further supplementation.

-The conclusions were based on the observed results which confirmed higher volumes of both ventricles when measured with manual vs threshold based tracings. For comparison with flow measurements the volumetric data showed closest correlation for both ventricles while obtained at k30 threshold which according the authors should be preferred to higher threshold as the substitute for manual segmentation. Moreover, the author found that the interobserver agreement was better for RV volumes rendered by threshold vs manual method. These all observation address the main aim of the study.

-The references are well chosen and appropriate.

-Figure 1 is clear and presents the idea of manual and threshold based calculations of RV and LV mass and volume. The addition of one-sentence-long comment concerning the chosen individual patient may make it more case-focused and illustrative by showing achieved values of presented parameters.

-Figures 2-1 and 2-1 are dedicated to Bland-Altman analysis and presented in typical way.

Tables are generally devoid of columns showing p valve (concerns Tables from 2 to 5) which is however acceptable while showing comparison of 4 or even five groups of data. The comments concerning p value are written in the text. 

Author Response

Thank you!

Reviewer 4 Report

This aritlce is well-written and clearly addressed the aim. 

The study retrospectively analyzed 94 patients biventricular functions from cine images of CMR study. These patients were referred for various cardiovascular diseases and the mean age was 15+/-9 years.

The study conclusded that manual and 30% threshold-based biventricular volume segmentation had best agreement with 2D, phanton-corrected PC flow measurements in young cardiac disease patients.

Major comments: 

1. The LVEF is generally normal, preserved or just mildly reduced. The moderate to severe LV systolic dysfunction can cause the stagnant/heterogenous flow in ventricle and flow artifacts. The limitation of threshold-based method is the quality of CMR images to delineate the myocardial TPM. Once the image quality is not good, we cannot manually revise it since the TPM cannot be seen clearly. In contrast, the manual method for segmentation of endocardial/epicardial border can still be roughly drawn and evaluate the biventricular function.

2. How about the patients with cardiac device? Since gredient echo is applied to cine images to minimize device artifact, the resolution is not as good as SSFP cine images.  

3. In the previous published paper ( doi: 10.1186/s12968-015-0111-7) for normal reference of LV/RV EDV index, ESV index, SV index, mass index, the myocardial mass can be or not to be calculated as part of LV/RV volume. Therefore, for individual report of LV/RV volume index, we can have compare to two references--with or without counting in TPM. There was only 13 healthy subjects in this study. We don't know the "normal reference" while using 30% threshold-based biventricular volume segmentation. Therefore, it's still need further investigation. Could authors comment on this viewpoint for clinical application of this method to evaluate cardiac functions? 

Minor  comments

Table 1:  please make the valubles arragement in good order.

Author Response

This aritlce is well-written and clearly addressed the aim. 

The study retrospectively analyzed 94 patients biventricular functions from cine images of CMR study. These patients were referred for various cardiovascular diseases and the mean age was 15+/-9 years.

The study conclusded that manual and 30% threshold-based biventricular volume segmentation had best agreement with 2D, phanton-corrected PC flow measurements in young cardiac disease patients.

Major comments: 

  1. The LVEF is generally normal, preserved or just mildly reduced. The moderate to severe LV systolic dysfunction can cause the stagnant/heterogenous flow in ventricle and flow artifacts. The limitation of threshold-based method is the quality of CMR images to delineate the myocardial TPM. Once the image quality is not good, we cannot manually revise it since the TPM cannot be seen clearly. In contrast, the manual method for segmentation of endocardial/epicardial border can still be roughly drawn and evaluate the biventricular function.

> That’s right, and image quality was good for both manual delineation of the endocardial and epicardial borders and for threshold-based segmentation – but even then, there were differences between the choices, which we showed in this article.

  1. How about the patients with cardiac device? Since gredient echo is applied to cine images to minimize device artifact, the resolution is not as good as SSFP cine images. 

>Patients with cardiac devices were not included, as we rarely scan those in our program. We added this condition now in the paragraph about exclusion criteria.

  1. In the previous published paper ( doi: 10.1186/s12968-015-0111-7) for normal reference of LV/RV EDV index, ESV index, SV index, mass index, the myocardial mass can be or not to be calculated as part of LV/RV volume. Therefore, for individual report of LV/RV volume index, we can have compare to two references--with or without counting in TPM. There was only 13 healthy subjects in this study. We don't know the "normal reference" while using 30% threshold-based biventricular volume segmentation. Therefore, it's still need further investigation. Could authors comment on this viewpoint for clinical application of this method to evaluate cardiac functions? 

> You are absolutely right. A sentence to this effect has now been added to the Discussion paragraph.

Minor  comments

Table 1:  please make the valubles arragement in good order.

> The formatting has now been improved, also for the other tables.

Round 2

Reviewer 2 Report

I have no further comments.

Reviewer 4 Report

I have no further comments.